# Randomized Experimental Design for Causal Graph Discovery

**Huining Hu**
School of Computer Science, McGill University.
huining.hu@mail.mcgill.ca

**Zhentao Li**
LIENS, École Normale Supérieure
zhentao.li@ens.fr

**Adrian Vetta**
Department of Mathematics and Statistics and School of Computer Science, McGill University.
vetta@math.mcgill.ca

## Abstract

We examine the number of controlled experiments required to discover a causal graph. Hauser and Buhlmann [1] showed that the number of experiments required is logarithmic in the cardinality of maximum undirected clique in the essential graph. Their lower bounds, however, assume that the experiment designer cannot use randomization in selecting the experiments. We show that significant improvements are possible with the aid of randomization – in an adversarial (worst-case) setting, the designer can then recover the causal graph using at most $O(\log \log n)$ experiments in expectation. This bound cannot be improved; we show it is tight for some causal graphs.

We then show that in a non-adversarial (average-case) setting, even larger improvements are possible: if the causal graph is chosen uniformly at random under a Erdös-Rényi model then the expected number of experiments to discover the causal graph is constant. Finally, we present computer simulations to complement our theoretic results.

Our work exploits a structural characterization of essential graphs by Andersson et al. [2]. Their characterization is based upon a set of orientation forcing operations. Our results show a distinction between which forcing operations are most important in worst-case and average-case settings.

## 1  Introduction

We are given $n$ random variables $V = \{V_1, V_2, \ldots, V_n\}$ and would like to learn the causal relations between these variables. Assume the dependencies between the variables can be represented as a directed acyclic graph $G = (V, A)$, known as the *causal graph*. In seminal work, Sprites, Glymour, and Scheines [3] present methods to obtain structural information on $G$ from passive observational data. In general, however, observational data can be used to discover only a part of the causal graph $G$; specifically, observation data will recover the *essential graph* $\mathcal{E}(G)$. To recover the entire causal graph $G$ we may undertake experiments. Here, an experiment is a controlled intervention on a subset $S$ of the variables. A controlled intervention allows us to deduce information about which variables $S$ influences.

The focus of this paper is to understand how many experiments are required to discover $G$. This line of research was initiated in a series of works by Eberhardt, Glymour, and Scheines (see [4, 5, 6]). First, they showed [4] that $n - 1$ experiments suffice when interventions can only be made upon singleton variables. For general experiments, they proved [5] that $\lceil \log n \rceil$ experiments are sufficient and, in the worst case necessary, to discover $G$. Eberhardt [7] then conjectured that $\lceil \log(\omega(G)) \rceil$

experiments are sufficient and, in the worst case, necessary; here $\omega(G)$ is the size of a maximum clique in $G$.[1] Hauser and Buhlmann [1] recently proved (a slight strengthening of) this conjecture. The essential mathematical concepts underlying this result can be traced back to work of Cai [8] on "separating systems" [9]; see also Hyttinen et al. [10].

Eberhardt [11] proposed the use of randomization (mixed strategies) in causal graph discovery. He proved that, if the designer is restricted to single-variable interventions, the worst case expected number of experiments required is $\Theta(n)$. Eberhardt [11] considered multi-variable interventions to be "far more complicated" to analyze, but hypothesized that $O(\log n)$ experiments may be sufficient, in that setting, in the worst-case.

## 1.1   Our Results

The purpose of this paper is to show that the lower bounds of [5] and [1] are not insurmountable. In essence, those lower bounds are based upon the causal graph being constructed by a powerful adversary. This adversary must pre-commit to the causal graph in advance but, before doing so, it has access to the entire list of experiments $\mathcal{S} = \{S_1, S_2, \dots\}$ that the experiment designer will use; here $S_i \subseteq V$ for all $i$. (This adversary also describes the "separating system" model of causal discovery. In Section 2.4 we will explain how this adversary can also be viewed as *adaptive*. The adversary may be given the list of experiments in order over time, but at time $i$ it needs only commit to the arcs in $\delta(S_i)$, the set of edges with exactly one end-vertex in $S_i$.)

Our first result is that we show this powerful adversary can be tricked if the experiment designer uses randomization in selecting the experiments. Specifically, suppose the designer selects the experiments $\{S_1, S_2, \dots\}$ from a collection of probability distributions $\mathcal{P} = \{\mathcal{P}_1, \mathcal{P}_2, \dots\}$, respectively, where distribution $\mathcal{P}_{i+1}$ may depend upon the results of experiments $1, 2 \dots, i$. Then, even if the adversary has access to the list of probability distributions $\mathcal{P}$ before it commits to the causal graph $G$, the expected number of experiments required to recover $G$ falls significantly. Specifically, if the designer uses randomization then, in the worst case, only at most $O(\log \log n)$ experiments in expectation are required. This result is given in Section 3, after we have presented the necessary background on causal graphs and experiments in Section 2. We also prove our lower bound is tight. This worst case result immediately extends to the case where the adversary is also allowed to use randomization in selecting the causal graph. Thus, the $O(\log \log n)$ bound applies to mixed-strategy equilibria in the game framework [11] where multi-variable interventions are allowed.

Our second result is that even more dramatic improvements are possible if the causal graph is non-adversarial. For a typical causal graph needs only a constant number of experiments are required in expectation! Specifically, if the directed acyclic graph is random, based upon an underlying Erdös-Rényi model, then $O(1)$ experiments in expectation are required to discover $G$. We prove this result in Section 4.

Our work exploits a structural characterization of essential graphs by Andersson et al. [2]. Their characterization is based upon a set of four operations. One operation is based upon acyclicity, the other three are based upon $v$-shapes. Our results show that the acyclicity operation is most important in improving worst-case bounds, but the $v$-shape operations are more important for average-case bounds. This conclusion is highlighted by our simulation results in Section 5. These simulations confirm that, by exploiting the $v$-shape operations, causal graph discovery is extremely quick in the non-adversarial setting. In fact, the constant in the $O(1)$ average-case guarantee may be even better than our theoretical results suggest. Typically, it takes one or two experiments to discover a causal graph on 15000 vertices!

## 2   Background

Suppose we want to discover an (unknown) directed acyclic graph $G = (V, A)$ and we are given its observational data. Without experimentation, we may not be able to recover all of $G$ from its observation data. But we can deduce a subgraph of it known as the *essential graph* $\mathcal{E}(G)$. In this section, we describe this process and explain how experiments (deterministic or randomized) can then be used to recover the rest of the graph. Throughout this paper, we assume the causal graph

and data distribution obey the faithfulness assumption and causal sufficiency [3]. The faithfulness assumption ensures that all independence relationships revealed by the data are results of the causal structure and are not due to some coincidental combinations of parameters. Causal sufficiency means there are no latent (that is, hidden) variables. These assumptions are important as they provide a one to one mapping between data and causal structure.

## 2.1 Observational Equivalence

First we may discover the *skeleton* and all the *v-structures* of $G$. To explain this, we begin with some definitions. The *skeleton of $G$* is the undirected graph on $V$ with an undirected edge (between the same endpoints) for each arc of $A$. A *v-shape* in a graph (directed or undirected) is an ordered set $(a, b, c)$ of three distinct vertices with exactly two edges (arcs), both incident to $b$. The *v-structures*, sometimes called *immoralities* [2], are the set of $v$-shapes $(a, b, c)$ where $ab$ and $cb$ are arcs. Two directed graphs with indistinguishable by observational data are said to belong to the same *Markov equivalence class*. Specifically, Verma and Pearl [12] and Frydenberg [13] showed the skeleton and the set of $v$-structures determine which equivalence class $G$ belongs to.

**Theorem 2.1.** *(Observational Equivalence) $G$ and $H$ are in the same Markov equivalence class if and only if they have the same skeletons and the same sets of $v$-structures.* □

Because of this equivalence, we will think of an observational Markov equivalence class as given by the skeleton and the set of (all) $v$-structures. From the observational data it is straightforward [12] to obtain the *basic graph* $\mathcal{B}(G)$, a mixed graph[2] obtained from the skeleton of $G$ by orienting the edges in each $v$-structure. For example, to test for an edge $\{i, j\}$, simply check there is no $d$-separator for $i$ and $j$; to test for a $v$-structure $(i, k, j)$, simply check that there is no $d$-separator for $i$ and $j$ that contains $k$. (These tests are not polynomial time. However, this is not relevant for the question we address in this paper.)

## 2.2 The Essential Graph

In fact, from the observational data we may orient more edges than simply those in the basic graph $\mathcal{B}(G)$. Specifically we can obtain the essential graph $\mathcal{E}(G)$. The *essential graph* is a mixed graph that also includes every edge orientation that is present in **every** directed acyclic graph that is compatible with the data. That is, an edge is oriented if and only if it has the same orientation in every graph in the equivalence class. For example, an edge $\{a, b\}$ is forced to be oriented as the arc $ab$ for the following reasons.

- $(F_1)$ The arc $ab$ (and the arc $cb$) is forced if it belongs to a $v$-structure $(a, b, c)$.
- $(F_2)$ There is a $v$-shape $(b, a, c)$ but it is not a $v$-structure. Then arc $ab$ is forced if $ca$ is an arc.
- $(F_3)$ The arc $ab$ is forced, by acyclicity, if there is already a directed path $P$ from $a$ to $b$.
- $(F_4)$ There is a $v$-shape $(c_1, a, c_2)$ but it is not a $v$-structure. Then the arc $ab$ is forced if there are directed paths $Q_1$ and $Q_2$ from $c_1$ to $b$ and from $c_2$ to $b$, respectively.

The reader can find illustrations of these forcing mechanisms in Figure 2 of the supplemental material. Andersson et al. [2] showed that these are the *only* ways to force an edge to become oriented. In fact, they characterize essential graphs and show only local versions of $(F_3)$ and $(F_4)$ are needed to obtain the essential graph – that is, it suffices to assume the path $P$ has two arcs and the paths $Q_1$ and $Q_2$ have only one arc each.

Let $\mathcal{U}(G)$ be the subgraph induced by the *undirected edges* of the essential graph $\mathcal{E}(G)$. For simplicity, we will generally just use the notation $\mathcal{B}, \mathcal{E}$ and $\mathcal{U}$. From the characterization, it can be shown that $\mathcal{U}$ is a chordal graph.[3] We remark that this chordality property is extremely useful in quantitatively analyzing the performance of the experiments we design. In particular, the size of the maximum clique and the chromatic number can be computed in linear time.

**Corollary 2.2.** *[2] The subgraph $\mathcal{U}$ is chordal.* □

## 2.3 Experimental Design

So observation data (the null experiment) will give us the essential graph $\mathcal{E}$. If we perform experiments then we may recover the entire causal graph $G$ and, in a series of works, Eberhardt, Glymour, and Scheines [5, 4, 6] investigated the number of experiments required to achieve this. An experiment is a controlled intervention that forces a distribution, chosen by the designer, on a set $S \subset V$. A key fact is that, given the existence of an edge $(a, b)$ in $G$, an experiment on $S$ can perform a *directional test* on $(a, b)$ if $(a, b) \in \delta(S)$ (that is, if exactly one endpoint of the edge is in $S$); see [5] for more details. Recall that we already know the skeleton of $G$ from the observational data. Thus, we can determine the existence of every edge in $G$. It then follows that to recover the entire causal graph it suffices that ($\Psi$) Each edge undergoes one directional test. The separating systems method is based on this sufficiency condition ($\Psi$). Using this condition, it is known that $\log n$ experiments suffice [5]. In fact, this bound can be improved to $\log \omega(\mathcal{U})$, where $\omega(\mathcal{U})$ is the size of the maximum clique in the undirected subgraph $\mathcal{U}$ of the essential graph $\mathcal{E}$. For completeness we show this result here; see also [8] and [1].

**Theorem 2.3.** *We can recover $G$ using $\log \omega(\mathcal{U})$ experiments.*

*Proof.* First use the observational data to obtain the skeleton of $G$. To find the orientation of each edge, take a vertex colouring $c : V(\mathcal{U}) \to \{0, 1, \dots, \chi(\mathcal{U}) - 1\}$, where $\chi(\mathcal{U})$ is the chromatic number of $\mathcal{U}$. We use this colouring to define our experiments. Specifically, for the $i$th experiment, select all vertices whose colour is 1 in the $i$th bit. That is, select $S_i = \{v : bin_i(c(v)) = 1\}$, where $bin_i$ extracts the $i$th bit of a number. Now, if vertices $u$ and $v$ are adjacent in $\mathcal{U}$, they receive different colours and consequently their colours differ at some bit $j$. Thus, in the $j$th experiment, one of $u, v$ is selected in $S_j$ and the other is not. This gives a directional test for the edge $\{u, v\}$. Therefore, from all the experiments we find the orientation of every edge. The result follows from the fact that chordal graphs are perfect (see, for example, [14]). $\square$

But ($\Psi$) is just a sufficiency condition for recovering the entire causal graph $G$; it need not be necessary to perform a directional test on every edge. Indeed, we may already know some edge orientations from the essential graph $\mathcal{E}$ via the forcing operations $(F_1), (F_2), (F_3)$ and $(F_4)$. Furthermore, the experiments we carry out will force some more edge orientations. But then we may again apply the forcing operations $(F_1)$-$(F_4)$ incorporating these new arcs to obtain even more orientations.

Let $\mathcal{S} = \{S_1, S_2, \dots S_k\}$, where $S_i \subseteq V$ for all $1 \le i \le k$, be a collection of experiments, Then the *experimental graph* is a mixed graph that includes every edge orientation that is present in **every** directed acyclic graph that is compatible with the data and the experiments $\mathcal{S}$. We denote the experimental graph by $\mathcal{E}_{\mathcal{S}}^+(G)$. Thus the question Eberhardt, Glymour, and Scheines pose is: how many experiments are needed to ensure that $\mathcal{E}_{\mathcal{S}}^+(G) = G$? As before, we know how to find the experimental graph.

**Theorem 2.4.** *The experimental graph $\mathcal{E}_{\mathcal{S}}^+(G)$ is obtained by repeatedly applying rules $(F_1)$–$(F_4)$ along with the rule:*
*$(F_0)$ There is an experiment $S_i \in \mathcal{S}$ and an edge $(a, b)$, with $a \in S_i$ and $b \notin S_i$. Then either the arc $ab$ or the arc $ba$ is forced depending upon the outcome of the experiment.* $\square$

We note that the proof uses the fact that arcs forced by $(F_0)$ are the union of edges across a set of cuts; without this property, a fourth forcing rule may be needed [15].

Theorem 2.4 suggests that it may be possible to improve upon the $\log \omega(\mathcal{U})$ upper bound. Unfortunately, Hauser and Buhlmann [1] show using an adversarial argument that in the worst case there is a matching lower bound, settling a conjecture of Eberhardt [6].

## 2.4 Randomized Experimental Design

As discussed in the introduction, the lower bounds of [5] and [1] are generated via a powerful adversary. The adversary must pre-commit to the causal graph in advance but, before doing so, it has access to the entire list of experiments $\mathcal{S} = \{S_1, S_2, \dots\}$ that the experiment designer will use. For example, assume that the adversary choses a clique for $G$ and the experiment designer selects a collection of experiments $\mathcal{S} = \{S_1, S_2, \dots\}$. Given the knowledge of $\mathcal{S}$ then, for a worst case performance, the adversary will direct every edge in $\delta(S_1)$ from $S_1$ to $V \setminus S_1$. The adversary will

then direct every edge in $\delta(S_2)$ (that has yet to be assigned an orientation) from $S_2$ to $V \setminus S_2$, etc. It is not difficult to show that the designer will need to implement at least $\log n$ of the experiments.

We remark that there is an alternative way to view the adversary. It need commit only to the essential graph in advance but otherwise may adaptively commit the rest of the graph over time. In particular, at time $i$, after experiment $S_i$ is conducted it must commit only to the arcs in $\delta(S_i)$ and to any induced forcings. This second adversary is clearly weaker than the first, but the lower bounds of [5] and [1] still apply here. Again, though, even this form of adversary appears unnaturally strong in the context of causal graphs. In particular, given the random variables $V$ the causal relations between them are pre-determined. They are already naturally present before the experimentation begins, and thus it seems appropriate to insist that the adversary pre commit to the graph rather than construct it adaptively.

Regardless, both of these adversaries can be countered if the designer uses randomization in selecting the experiments. In particular, in randomized experimental design we allow the designer to select the experiments $\{S_1, S_2, \dots\}$ from a collection of probability distributions $\mathcal{P} = \{\mathcal{P}_1, \mathcal{P}_2, \dots\}$, respectively, where distribution $\mathcal{P}_{i+1}$ may depend upon the results of experiments $1, 2 \dots, i$. As an example, consider again the case in which the adversary selects a clique. Suppose now that the designer selects the first experiment $S_1$ uniformly at random from the collection of subsets of cardinality $\frac{1}{2}n$. Even given this knowledge, it is less obvious how the adversary should act against such a design. Indeed, in this article we show the usefulness of the randomized approach. It will allow the designer to require only $O(\log \log n)$ experiments in expectation. This is the case even if the adversary has access to the entire list of probability distributions $\mathcal{P}$ before it commits to the causal graph $G$. We prove this in Section 3. Thus, by Theorem 2.3, we have that $\min[O(\log \log n), \log \omega(\mathcal{U})]$ experiments are sufficient. We also prove that this bound is tight; there are graphs for which $\min[O(\log \log n), \log \omega(\mathcal{U})]$ experiments are necessary.

Still our new lower bound only applies to causal graphs selected adversarially. For a typical causal graph we can do even better. Specifically, we prove, in Section 4, that for a random causal graph a constant number of experiments is sufficient in expectation. Consequently, for a random causal graph the number of experiments required is independent of the number of vertices in the graph! This surprising result is confirmed by our simulations. For various values $n$ of number of vertices, we construct numerous random causal graphs and compute the average and maximum number of experiments needed to discover them. Simulations confirm this number does not increase with $n$.

Our results can be viewed in the game theoretic framework of Eberhardt [11], where the adversary selects a probability distribution (mixed strategy) over causal graphs and the experiment designer choses a distribution over which experiments to run. In this zero-sum game, the payoff to the designer is the negative of the number of experiments needed. The worst case setting corresponds to the situation where the adversary can choose any distribution over causal graphs. Thus, our result implies a worst case $-\Theta(\log \log n)$ bound on the value of a game with multi-variable interventions and no latent variables. Therefore, the ability to randomize turns out to be much more helpful to the designer than the adversary. Our average case $O(1)$ bound corresponds to the situation where the adversary in the game is restricted to choose the uniform distribution over causal graphs.

## 3 Randomized Experimental Design

### 3.1 Improving the Upper Bound by Exploiting Acyclicity

We now show randomization significantly reduces the number of experiments required to find the causal graph. To improve upon the $\log \chi(\mathcal{U})$ bound, recall that $(\Psi)$ is a sufficient but not necessary condition. In fact, we will not need to apply directional tests to every edge. Given some edge orientations we may obtain other orientations for free by acyclicity or by exploiting the characterization of [2]. Here we show that the acyclicity forcing operation $(F_3)$ on its own provides for significant speed-ups when we allow randomisation.

**Theorem 3.1.** *To orient a clique on $t$ vertices, $O(\log \log t)$ experiments suffice in expectation.*

*Proof.* Let $\{x_1, x_2, \dots, x_t\}$ be the true acyclic ordering of the clique $G$. Now take a random experiment $S$, where each vertex is independently selected in $S$ with probability $\frac{1}{2}$. The experiment $S$ partitions the ordering into *runs (streaks)* – contiguous segments of $\{x_1, x_2, \dots, x_t\}$ where either

every vertex of the segment is in $S$ or every vertex of the segment is in $\bar{S} = V \setminus S$. Without loss of generality the first run is in $S$ and we denote it by $R_0$. We denote the second run, which is in $\bar{S}$, by $\bar{R}_0$, the third run by $R_1$, the fourth run by $\bar{R}_1$ etc. A well known fact (see, for example, [16]) is that, with high probability, the longest run has length $\Theta(\log t)$.

Take any pair of vertices $u$ and $v$. We claim that edge $\{u, v\}$ can be oriented provided the two vertices are in different runs. To see this first observe that the experiment will orient any edge between $S$ and $\bar{S}$. Thus if $u \in R_i$ and $v \in \bar{R}_j$, or vice versa, then we may orient $\{u, v\}$. Assume $u \in R_i$ and $v \in R_j$, where $i < j$. We know $\{i, j\}$ must be the arc $ij$, but how do we conclude this from our experiment? Well, take any vertex $w \in \bar{R}_i$. Because $G$ is a clique there are edges $\{u, w\}$ and $\{v, w\}$. But these edges have already been oriented as $uw$ and $wv$ by the experiment. Thus, by acyclicity the arc $uv$ is forced. A similar argument applies for $u \in \bar{R}_i$ and $v \in \bar{R}_j$, where $i < j$.

It follows that the only edges that cannot be oriented lie between vertices within the same run. Each run induces an undirected clique after the experiment, but each such clique has cardinality $O(\log t)$ with high probability. We can now independently and simultaneously apply the deterministic method of Theorem 2.3 to orient the edges in each of these cliques using $O(\log \log t)$ experiments. Hence the entire graph is oriented using $1 + O(\log \log t)$ experiments. $\qquad\square$

We note that if any high probability event does not occur, we simply restart with new random variables, at most doubling the number of experiments (and tripling if it happens again and so on). The expected number of experiments is then the number we get with no restart multiplied by $\sum_i ip^i$, which is bounded by a constant (usually approaching 1 if $p$ is a decreasing function of $t$).

Theorem 3.1 applies to cliques. The same guarantee, however, can be obtained for any graph.

**Theorem 3.2.** *To construct $G$, $O(\log \log n)$ experiments suffice in expectation.*

*Proof.* Take any graph $G$ with $n$ vertices. Recall, we only need orient the edges of the chordal graph $\mathcal{U}$. But a chordal graph contains at most $n$ maximal cliques [14] (each of size $t \le n$). Suppose we perform the randomized experiment where each vertex is independently selected in $S$ with probability $\frac{1}{2}$, as in Theorem 3.1. Then any vertex of a maximal clique $Q$ is in $S$ with probability $\frac{1}{2}$. Thus, this experiment breaks $Q$ into runs all of cardinality at most $O(\log n)$ with high probability.[4] Since there are only $n$ maximal cliques, applying the union bound gives that every maximal clique in $\mathcal{U}$ is broken up into runs of cardinality $O(\log n)$ with high probability. Therefore, since every clique is a subgraph of a maximal clique, after a single randomized experiment, the chordal graph $\mathcal{U}'$ formed by the remaining undirected edges has $\omega = O(\log n)$. We can now independently apply Theorem 2.3 on $\mathcal{U}'$ to orient the remaining edges using $O(\log \log n)$ experiments. $\qquad\square$

We can also iteratively exploit the essential graph characterization [2] but in the worst case we will have no $v$-structures and so the expected bound above will not be improved. Combining Theorem 2.3 and Theorem 3.2 we obtain

**Corollary 3.3.** *To construct $G$, $\min[O(\log \log n), \log \omega(\mathcal{U})]$ experiments suffice in expectation.* $\quad\square$

## 3.2 A Matching Lower Bound

The bound in Corollary 3.3 cannot be improved. In particular, the bound is tight for unions of disjoint cliques. (Due to space constraints, this proof is given in the supplemental materials.)

**Lemma 3.4.** *If $G$ is a union of disjoint cliques, $\Omega\left(\min[\log \log n, \log \omega(\mathcal{U})]\right)$ experiments are necessary in expectation to construct $G$.*

Observe that Lemma 3.4 explains why attempting to recursively partition the runs (used in Theorem 3.1) in sub-runs will not improve worst-case performance. Specifically, a recursive procedure may produce a large number of sub-runs and, with high probability, the trick will fail on one of them.

# 4   Random Causal Graphs

In this section, we go beyond worst-case analysis and consider the number of experiments needed to recover a typical causal graph. To do this, however, we must provide a model for generating a "typical" causal graph. For this task, we use the Erdös-Rényi (E-R) random graph model. Under this model, we show that the expected number of experiments required to discover the causal graph is just a constant. We remark that we chose the E-R model because it is the predominant graph sampling model. We do not claim that the E-R model is the most appropriate random model for every causal graph application. However, we believe the main conclusion we draw, that the expected number of experiments to orient a typical graph is very small, applies much more generally. This is because the vast improvement we obtain for our average-case analysis (over worst-case analysis) is derived from the fact that the E-R model produces many $v$-shapes. Since any other realistic random graph model will also produce numerous $v$-shapes (or small clique number), the number of experiments required should also be small in those models.

Now, recall that the standard Erdös-Rényi random graph model generates an undirected graph. The model, though, extends naturally to directed, acyclic graphs as well. Specifically, our graphs $C_{n,p}$ with parameters $n$ and $p$ are chosen according to the following distribution:
(1) Pick a random permutation $\sigma$ of $n$ vertices.
(2) Pick an edge $(i, j)$ (with $1 \leq i < j \leq n$) independently with probability $p$.
(3) If $(i, j)$ is picked, orient it from $i$ to $j$ if $\sigma(i) < \sigma(j)$ and from $j$ to $i$ otherwise.

Note that since each edge was chosen randomly, we obtain the same distribution of causal graphs if we simply fix $\sigma$ to be the identity permutation. In other words, $C_{n,p}$ is just a random undirected graph $G_{n,p}$ in which we've directed all edges from lower to higher indexed vertices. Clearly, this graph is then acyclic. The main result in this section is that the expected number of experiments needed to recover the graph is constant. We prove this in the supplemental materials.

**Theorem 4.1.** *For $p \leq \frac{4}{5}$ we can recover $C_{n,p}$ using at most $\log \log 13$ experiments in expectation.*

We remark that the probability $\frac{4}{5}$ in Theorem 4.1 can easily be replaced by $1 - \delta$, for any $\delta > 0$. The resulting expected number of experiments is a constant depending upon $\delta$. Note, also, that the result holds even if $\delta$ is a function of $n$ tending to zero. Furthermore, we did not attempt to optimize the constant $\log \log 13$ in this bound.

Theorem 4.1 illustrates an important distinction between worst-case and average-case analyses. Specifically, the bad examples for the worst-case setting are based upon clique-like structures. Cliques have no $v$-shapes, so to improve upon existing results we had to exploit the acyclicity operation $(F_3)$. In contrast, for the average-case, the proof of Theorem 4.1 exploits the $v$-structure operation $(F_1)$. The simulations in Section 5 reinforce this point: in practice, the operations $(F_1, F_2, F_4)$ are extremely important as $v$-shapes are likely to arise in typical causal graphs.

# 5   Simulation Results

In this section, we describe the simulations we conducted in MATLAB. The results confirm the theoretical upper bounds of Theorem 4.1; indeed the results suggest that the expected number of experiments required may be even smaller than the constant produced in Theorem 4.1. For example, even in graphs with 15000 vertices, the average cardinality of the maximum clique in the simulations is only just over two! This suggests that the full power of the forcing rules $(F_1)$-$(F_4)$ has not been completely measured by the theoretical results we presented in Sections 3 and 4.

For the simulations, we first generate a random causal graph $G$ in the E-R model. We then calculate the essential graph $\mathcal{E}(G)$. To do this we apply the forcing rules $(F_1)$-$(F_4)$ from the characterization of [2]. At this point we examine properties of the $\mathcal{U}(G)$ the undirected subgraph of $\mathcal{E}(G)$. We are particularly interested in the maximum clique size in $\mathcal{U}$ because this information is sufficient to upper bound the number of experiments that **any** reasonable algorithm will require to discover $G$.

We remark that, to speed up the simulations we represent a random graph $G$ by a symmetric adjacency matrix $M$. Here, if $M_{i,j} = 1$ then there is an arc $ij$ if $i < j$ and an arc $ji$ if $i > j$. The matrix formulation allows the forcing rules $(F_1)$-$(F_4)$ to be implemented more quickly than standard approaches. For example, the natural way to apply the forcing rule $(F_1)$ is to search explicitly for each $v$-structure of which there may be $O(n^3)$. Instead we can find every edge contained in a $v$-structure

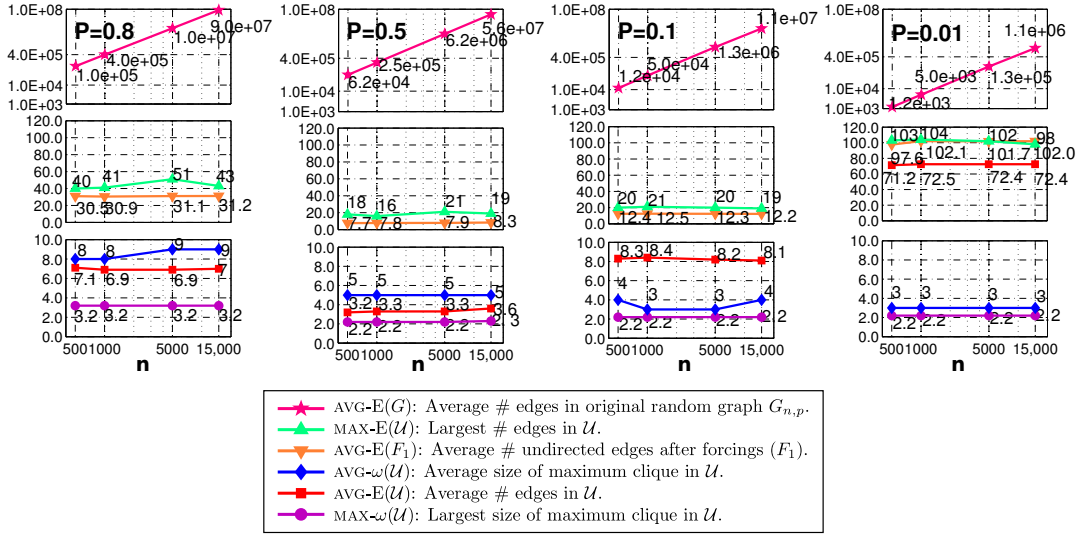

Figure 1: Experimental results: number of edges and size of the maximum cliques for $C_{n,p}$

using matrix multiplication, which is fast under MATLAB.[5] The validity of such an approach can be seen by the following theorem whose proof is left to the supplemental material.

**Theorem 5.1.** *Given the adjacency matrix $M$ of a causal graph, we can find all edges contained in a $v$-structure via matrix multiplication.*

To speed up computation for smaller values of $p$ and large $n$, we instead used sparse matrices to apply $(F_1)$ storing only a list of non-zero entries ordered by row and column and vice versa. Then matrix multiplication could be performed quickly by looking for common entries in two short lists.

We ran simulations for four choices of probability $p$, specifically $p \in \{0.8, 0.5, 0.1, 0.01\}$, and for four choices of graph size $n$, specifically $n \in \{500, 1000, 5000, 15000\}$. For each combination pair $\{n, p\}$ we ran $1000$ simulations. For each random graph $G$, once no more forcing rules can be applied we have obtained the essential graph $\mathcal{E}(G)$. We then calculate $|E(\mathcal{U})|$ and $\omega(\mathcal{U})$. Our results are summarized in Figure 1.

Here average/largest refers to the average/largest over all $1000$ simulations for that $\{n, p\}$ combination. Observe that the lines for AVG-E($G$) and AVG-E($F_1$) illustrate Theorem 4.1: there is a dramatic fall in the expected number of undirected edges remaining by just applying the $v$-structure forcing operation $(F_1)$. The AVG-E($\mathcal{U}$) and MAX-E($\mathcal{U}$) show that the number of edges fall even more when we apply all the forcing operations to obtain $\mathcal{U}$.

More remarkably the maximum clique size in $\mathcal{U}$ is tiny, AVG-$\omega(\mathcal{U})$ is just around two or three for all our choices of $p \in \{0.8, 0.5, 0.1, 0.01\}$. The largest clique size we ever encountered was just nine. Since the number of experiments required is at most logarithmic in the maximum clique size, none of our simulations would ever require more than five experiments to recover the causal graph and nearly always required just one or two. Thus, the expected clique size (and hence number of experiments) required appears even smaller than the constant 13 produced in Theorem 4.1.

We emphasize that the simulations do not require the use of a specific algorithm, such as the algorithms associated with the proofs of the worst-case bound (Theorem 3.2) and the average-case bound (Theorem 4.1). In particular, the simulations show that the null experiment applied in conjunction with the forcing operations $(F_1)$-$(F_4)$ is typically sufficient to discover most of the causal graph. Since the remaining unoriented edges $\mathcal{U}$ have small maximum clique size, any reasonable algorithm will then be able to orient the rest of the graph using a constant number of experiments.

**Acknowledgement** We would like to thank the anonymous referees for their remarks that helped us improve this paper.

## Footnotes

[1]A directed graph is a *clique* if its underlying undirected graph is a (undirected) clique.

[2]A mixed graph contains oriented edges and unoriented edges. To avoid confusion, we refer to oriented edges as arcs.

[3]A graph $H$ is chordal if every induced cycle in $H$ contains exactly three vertices. That is, every cycle $C$ on at least four vertices has a *chord*, an edge not in $C$ that connects two vertices of the cycle.

[4]Specifically, every run will have cardinality at most $k \cdot \log n$ with probability at least $1 - \frac{1}{n^{k-1}}$.

[5]In theory, matrix multiplication can be carried in time $O(n^{2.38})$ [17].

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
