[Supplementary Material]

## Supplemental Material

**Illustrations of the Forcing Operations.**

Figure 2: Forced Orientations.

**Proof of Theorem 4.1.**

Without loss of generality, we may assume the vertex ordering is $\{1, 2, \ldots, n\}$. Given this ordering, we want to know what is the probability that vertex $t$ is the $r$th (smallest) vertex in a undirected clique in $\mathcal{U}$. Observe that there are $\binom{t-1}{r-1}$ possible $r$-cliques ending at vertex $t$. Take any such clique $Q$. Let $A_Q$ be the indicator variable for $Q$ to be a clique in $\mathcal{U}$.

To calculate the probability of this event, first note that each edge $e = (j, i), j < i \leq t$ in $Q$ must have been selected in the random graph $C_{n,p}$. This occurs with probability $p^{\binom{r}{2}}$. Furthermore, no edge $(j, t) \in Q$ can be contained in any $v$-structure; otherwise its orientation will already have been discovered and it will not be in $\mathcal{U}$. For these events to occur, $(j, t)$ must be chosen as an edge in the random graph **and** for all $k < t, k \notin Q$, either $(k, t)$ is not an edge or $(k, t)$ is an edge and $(k, q)$ is an edge for all $q \in Q, q < t$. Thus, if $t$ is the $r$th smallest vertex in $Q$,

$$P(A_Q) \leq p^{\binom{r}{2}} \cdot \left((1-p) + p \cdot p^{r-1}\right)^{t-r}$$

Therefore, by the union bound, the probability that any $r$-clique ends at vertex $t$ is at most

$$\binom{t-1}{r-1} \cdot p^{\binom{r}{2}} \cdot (1-p+p^r)^{t-r} \leq \binom{t}{r} \cdot p^{\binom{r}{2}} \cdot (1-p+p^r)^{t-r}$$