[Reviews · NeurIPS 2014]

Submitted by Assigned_Reviewer_18

This paper is motivated by the following question. Suppose the causal structure is properly represented by a DAG over the given variables, and the essential graph that represents the Markov equivalence class of the true causal DAG has been learned from the observational data. Assuming that an experiment can actively control any subset of the given variables, and is ideally effective in deciding the direction of an arrow between a variable under control and a variable not under control, how many experiments are needed to fully determine the causal DAG from the essential graph? Previous work on this question (due mainly to Eberhardt et al. and Hauser and Buhlmann) did not quite address the situtation where the targets of the experiments can be randomized. This paper fills in the gap and establishes two main results on this problem: first, in the worst case, the required number of experiments is O(loglog n) with high probability (where n is the number of variables); second, in the average case (where the causal DAG is assumed to be generated from a DAG version of the Erdos-Renyi random graph model), the required number of experiments is bounded by a constant in expectation (for the expected maximal clique size is bounded by a constant, which is also illustrated by some simulations).

The results are novel and interesting, and the proofs look sound to me (though I regret to say that I didn’t have time to check them carefully). The technical details take some time to digest, but the main ideas of the arguments are sufficiently clear. I have a few minor comments:

1. The last sentence in the abstract says: “it shows that huge performance improvements are possible when moving from single-variable interventions to multi-variable interventions.” This remark does not seem to be picked up in the main text. Also, the general point seems to have been made in previous work.

2. I suspect that Theorem 2.4 is false. The task of constructing the “experimental essential graph” seems to me closely related to the task of constructing the essential graph given some background knowledge. As Meek (1995, “Causal Inference and Causal Explanation with Background Knowledge”) showed, an extra rule is sometimes needed to construct the essential graph when there is background knowledge.

UPDATE: Thanks to the author(s)'s clarification in the feedback, I now see that Meek's additional rule is not needed as the "background knowledge" obtained from experiments is appropriately restricted.

3. The paper seems to use “with high probability” and “in expectation” interchangeably. I would appreciate a clarification, as in my understanding, the two phrases do not have the exact same meaning.

4. Line 3 in Section 3.1, “adjacency tests” should be “directional tests”.

5. In the paragraph below Theorem 4.1, I suspect that the “p” in “Note, also, that the result holds even if p is a function of n tending to zero” is meant to be “\delta”. Also, for the said result, what does the constant bound depend on? The rate of convergence?

6. The proof of Theorem 4.1 shows that the expected maximal clique size is bounded by 13. Doesn’t this imply that the number of experiments stated in Theorem 4.1 should be log13?
Summary: Overall this is a readable paper with interesting and potentially useful results related to the active learning of causal graphs.

Submitted by Assigned_Reviewer_24

UPDATE AFTER DISCUSSION ROUND WITH AUTHORS:

thanks for accepting discussing your paper with us. Really liked reviewing it !
Still unclear to me when you state "Specifically, this result show that having many large cliques is not an insurmountable barrier." I would have thought that it is ?! See K. S. Sesh Kumar and F. Bach. Convex Relaxations for Learning Bounded Treewidth Decomposable Graphs. Technical report, HAL 00763921, 2012. Proceedings of the International Conference on Machine Learning (ICML) on the subject. A discussion in relationship to tree-width (in a future version of your paper) would really be a plus ! My understanding is that ONE dense and large clique is already a big problem...But in my framework, it's about structure network recovery, not to have an orientation from a graph skeleton...
Best wishes !

%%%%%%%%%%%%%%%%%%%%%%%%%%%%%%%%%%%%%%%%%%%%%%%%%%%%
This paper proposes to give conditions on the needed number of experiments to reconstruct a '"causal" graph from (interventional) data (given the skeleton is known from observational data). It follows from work of others ([1]). It states that only $ \log \log n $ experiments ($ n $ being the numbe of variables or nodes) are needed to recover the full causal structure in a "very adverse" situation (optimality of the result claimed) and this could fall down to a constant sample size when the graph is thought to be Erdös (is it really useful ??).
Section 1 is an introduction, Section 2 gives background on graph, causality and the potential to reconstruct from observations and single perturbations. Section 3 introduces multiple random experiments so as to improve the actually needed number of experiments to be ale to decipher the causal structure. Section 4 gives a stringer result in a more constrained situation where graphs are generated according to an Erdös-Renyi model. Section 5 presents some simulation results

Original results are clearly presented and seem to be an interesting step forward in a very important field: deciphering true causal relationships in complex systems and not only co-occurrence relationships. The paper is ok to read but some slight improvements could be made in the writing? A careful reading and the authors could be trusted to achieve a better readability !

Few comments:
- is it ok to put references in the astract in NIPS policy ? And to use CR when writing it ?
- l24: yes, I remember a presentation on Rau et al. BMC Syst. Biol. which also mentioned that there are much cleverer perturbation than single KO experiments ?! This makes sense and can be related to the graph structure at hand.
- last sentence: rephrase, this is unclear.
- l37: dependences or dependencies ?
- in [article missing] seminal work.
- l43: sentence could be turned better.
- try to avoid footnote. And footnote #1 is not a very accurate statement ?!
- Some work of Guyon and colleagues may be cited.
- why is "our results" section 1.1 and no 1.2 follows ?
- I assume the 'adversary' notion is clear for those familiar with the game framework of [2] but not for the entire NIPS readership ?!
- l61 "pre-commit" ??
- how do you framework compare to that of Peters & Bühlmann, Biometrika 2014 ?
- would you say that your result on Erdös-Renyi graph is surprising ? I would say NO, their structure is relatively poor as compared to simulated but slighlty richer classes of models. Not to speak about true graph which obviously fall in none of these simulated framework (Erdös-Renyi, scale-free, small world...).
- Table 1: I really beg you to avoid such a table ot present your results, they are difficult to read. Not on slides please ??!! And give a caption which has a signification please !
- A wee conclusion/discussion might be a plus...
Summary: An interesting paper on a challenging field with results that seem to be of great interest.
Some improvements in the writing welcome though.

Submitted by Assigned_Reviewer_33

This paper provides a tighter bound on the minimum number of experiments to infer a causal graph, by using a randomized experimental design. The paper is well written and builds up on previous works on this subject that did not exploit using randomization. The provided bound, as other bounds previously derived from the literature, depend on the sparsity pattern of the underlying causal graph, particularly on the maximum clique size. The authors carry out their simulation experiments using an Erdos-Renyi model acknowledging it may not be the most appropriate model but, simultaneously, claiming that they "believe" still that their result (the very small number of required experiments) holds. This is quite speculative and this reviewer wonders why the authors did not make an effort in using other random graph sampling models to give some empirical evidence of that claim. One obvious choice could be d-regular graphs (see http://en.wikipedia.org/wiki/Regular_graph) whose density/sparsity is a linear function of the constant vertex degree.
Summary: Good paper, clearly written.
Author Feedback
Author rebuttal: We thank the reviewers and accept their suggestions regarding the format and presentation.
We will now address the questions raised in the reviews.
(Due to space constraints, we will focus only upon the major issues raised.)

To All Reviewers:

The Erdos-Renyi model: Some reviewers have asked us why our simulations were run on this model.
One reason is that the graphs generated by the chosen model must satisfy certain properties, such
as being directed, acyclic and ideally that their v-structures are quick to compute.
Furthermore, we required a model that is potentially difficult for graph discovery.
Specifically, we need a model that naturally generates large cliques --
graphs with small maximum cliques are oriented quickly.
The Erdos-Renyi model is very well known and has this property.
We would gladly run simulations on other suitable models in future work.

To Reviewer_18:

2. This is a very interesting remark. Thank you for bringing it to our attention. To wit, the subtle
differences in the models allow both Theorem 2.4 and the result of Meek to hold.

Theorem 2.4 is closer to question (D), which Meek shows is possible to answer. The difference is that
in our case, the "background information" we can have (i.e., the results of our experiment) is
restricted in 3 ways.

(i) we have only required edges (i.e., no forbidden edges),
(ii) required edges are the union of edges across some set of cuts (as opposed to having
individual directed edges) and
(iii) that this background can always be completed as they always correspond the directed
edges of some true causal graph.

In Meek's model, this prior knowledge could potentially contradict observed data (in which case,
there is no causal graph consistent with data and prior knowledge). In our case, the answer to
question (A) and (B) of Meek would always be "yes" as it is simply one of our assumptions.

Rule R4 is an interesting one. However, because of (ii), it is not possible to obtain only those
directed edges as a result of experiments! In fact, we can show that the essential graph contains
no 3-cycles with some edges oriented and others unoriented after applying the other forcing rules.

We will add a proof of this in the full version.

3. Indeed, "with high probability" and "in expectation" do have different meanings.
However, in our setting, since the maximum number of experiments ever required is
not too large, a "with high probability" guarantee easily extends to a
similar "in expectation" guarantee. We will clarify this in the paper.

To Reviewer_24:

- "Would you say that your result on Erdos-Renyi graph is surprising ? I would say NO..."

Whilst an improvement was expected, the degree of improvement is perhaps more than
we anticipated. For example, for p=0.5 the ER model will typically induce a super-polynomial
number of cliques of size greater than sqrt(log n). That all these cliques are oriented
in O(1) experiments is a bit surprising.
Whether the result extends to most other models is open, but there are reasons to conjecture it does.
Specifically, this result show that having many large cliques is not an insurmountable barrier.

- "I assume the 'adversary' notion is clear for those familiar with the game framework of [2] but not
for the entire NIPS readership".

Adversary is the usual adversary for worst case analysis of algorithms. In [2], the game theoretic
framework was introduced and, since it is relevant past work, we had to introduce it here in order to
describe its result for comparison. We will work to make this distinction clearer.

- "pre-commit"

Indeed, we mean the adversary must choose a causal graph (with the vertices labelled by variables) before
the first experiment takes place.

- "How does your framework compare to that of Peters & Buhlmann?"

Peters and Buhlmann focus on the model where the random variables are related by linear equations with an error term.
In our work, we are only concerned with dependence amongst variables and make no assumptions as to what this dependence is.
In their case, when the error variances are the same, they may infer the graph from only observational data without
any experiments. In our case, we show a (matching) lower bound on the number of experiments needed in Section 3.2

To Reviewer_33:

- "One obvious choice could be d-regular graphs".

We are now aware of models that generate random d-regular DAGs.
Regardless, structurally they would be very similar to ER graphs with p=d/2n. So the O(1)
bound should still apply.